# Orthogonal Contrastive Learning for Multi-Representation fMRI Analysis

**Tony Muhammad Yousefnezhad[1,2],***
[1]Learning By Machine
[2]Information Management, National Bank of Canada
Edmonton AB Canada
tony@learningbymachine.com

## Abstract

Task-based functional magnetic resonance imaging (fMRI) provides invaluable insights into human cognition but faces critical hurdles—low signal-to-noise ratio, high dimensionality, limited sample sizes, and costly data acquisition—that are amplified when integrating datasets across subjects or sites. This paper introduces orthogonal contrastive learning (OCL), a unified multi-representation framework for multi-subject fMRI analysis that aligns neural responses without requiring temporal preprocessing or uniform time-series lengths across subjects or sites. OCL employs two identical encoders: an online network trained with a contrastive loss that pulls together same-stimulus responses and pushes apart different-stimulus responses, and a target network whose weights track the online network via exponential moving average to stabilize learning. Each OCL network layer combines QR decomposition for orthogonal feature extraction, locality-sensitive hashing (LSH) to produce compact subject-specific signatures, positional encoding to embed temporal structure alongside spatial features, and a transformer encoder to generate discriminative, stimulus-aligned embeddings. We further enhance OCL with an unsupervised pretraining stage on fMRI-like synthetic data and demonstrate a transfer-learning workflow for multi-site studies. Across extensive experiments on multi-subject and multi-site fMRI benchmarks, OCL consistently outperforms state-of-the-art alignment and analysis methods in both representation quality and downstream classification accuracy.

## 1 Introduction

Task-based functional magnetic resonance imaging (fMRI) is a widely used technique in neuroscience for studying brain activity during cognitive processes such as decision-making, perception, and attention [1, 2, 3, 4]. By capturing blood-oxygen-level-dependent (BOLD) signals while subjects engage in structured tasks, fMRI enables researchers to link brain regions to specific mental functions [5]. Despite its potential, fMRI data present several challenges: they are high-dimensional, inherently noisy, expensive to acquire, and often limited in sample size—factors that hinder the training and generalization of machine learning models [1, 2, 3, 4, 6, 7]. To mitigate these limitations, modern research increasingly relies on multi-subject fMRI datasets to improve model robustness and validity. Moreover, the growing availability of large-scale, open-access repositories such as the national

---

*This research study is conducted independently and is not connected to the author's role at the *National Bank of Canada.*

institute of mental health (NIMH) [2], the Human Connectome Project [3], and OpenNEURO [4] has made it feasible to aggregate homogeneous task-based fMRI data across multiple sites, thereby increasing sample diversity and statistical power [1, 7]. However, this introduces additional complexity, including inter-subject variability, cross-site differences in scanner hardware and acquisition protocols, and population-level heterogeneity [1, 7, 8]. Consequently, there is a pressing need for machine learning frameworks that can generalize across sites and subjects while being resilient to such batch effects, making the development of domain-adaptive, multi-representation learning techniques essential for real-world fMRI analysis.

Multi-subject fMRI analysis is complicated by substantial inter-individual variability in brain connectivity, as each person's connectome exhibits unique structural and functional patterns that lead to idiosyncratic neural responses across subjects [2, 9]. To address this, functional alignment techniques—most notably hyperalignment [5] and shared response model (SRM) [6, 10]—project each subject's neural responses into a shared representational space using an orthogonal mapping procedure, effectively realigning neural signatures and improving inter-subject correspondence. These alignment strategies can be framed as multi-view learning problems—each subject constitutes a 'view', and methods like generalized canonical correlation analysis (CCA) identify transformations that maximize shared information across views [2, 6, 9, 10, 11, 12, 13]. Recent work has extended functional alignment to multi-site fMRI studies, aiming to pool data from different scanners and populations; however, these efforts must contend with batch effects arising from scanner hardware differences, acquisition protocols, and site-specific demographics [1, 7, 8]. Such batch effects introduce unwanted variability that can confound downstream analyses unless corrected by harmonization methods such as domain-adaptation frameworks tailored to neuroimaging [1, 7]. Constructive learning [14, 15, 16, 17, 18, 19, 20] is a paradigm in which models learn by contrasting similar and dissimilar example pairs to shape feature spaces. It complements multi-view functional alignment by using contrastive objectives to directly align representations across subjects and sites and, by enforcing agreement on same-stimulus responses while discouraging spurious correlations, helps mitigate batch effects and enhances robustness and generalization in multi-site fMRI analyses.

The main contributions of this paper are fivefold: (1) we introduce orthogonal contrastive learning (OCL), a unified multi-representation framework that aligns multi-subject fMRI data without temporal preprocessing or uniform time-series length requirements across subjects or sites; (2) we design a dual-encoder architecture—an online network trained with a contrastive loss that pulls same-stimulus responses together and pushes different-stimulus responses apart, and a target network updated via exponential moving average to stabilize learning and enforce consistency; (3) we develop a novel OCL layer composed of four tightly integrated components: QR decomposition, which yields orthonormal feature bases to decorrelate signals and enhance the signal-to-noise ratio; locality-sensitive hashing (LSH) [21], which produces compact subject-specific signatures that preserve similarity relationships while drastically reducing feature dimensionality; positional encoding, which injects continuous temporal context into spatial feature representations to maintain dynamic stimulus information; and a transformer encoder, which employs multi-head self-attention to capture global dependencies and produce discriminative, stimulus-aligned embeddings; (4) we propose an unsupervised pretraining strategy on synthetic fMRI-like data to initialize OCL parameters for faster convergence and improved robustness; and (5) we demonstrate a transfer-learning pipeline that applies trained OCL models to multi-site datasets, showing resilience to scanner variability and sequence-length differences, and achieving superior downstream classification performance over state-of-the-art methods.

The remainder of this paper is structured as follows. Section 2 reviews related work, Section 3 details our proposed method, Section 4 presents our empirical evaluation, and Section 5 concludes with key findings and avenues for future research.

## 2 Related Works

Hyperalignment (HA) is a deterministic alignment technique that uses generalized CCA to enhance prediction accuracy in fMRI analysis [5, 11, 12]. Classic HA's requirement to invert high-dimensional covariance matrices makes it unreliable for highly correlated data—*e.g.*, whole-brain

---

[2]Available at `https://data-archive.nimh.nih.gov/`

[3]Available at `https://www.humanconnectome.org/`

[4]Available at `https://openneuro.org/`

images; variants such as regularized hyperalignment (RHA) [11], singular value decomposition hyperalignment (SVDHA) [22], and (non-parametric) kernel hyperalignment (KHA) [12] respectively introduce regularization, low-rank decompositions, or kernel mappings to stabilize alignment. Deep hyperalignment (DHA) further extends this line by employing a deep neural network as a learnable kernel to capture complex, nonlinear subject-specific transformations end-to-end [9]. More recently, deep geodesic canonical correlation analysis (DeepGeoCCA) has been proposed to generalize CCA to symmetric positive-definite covariance matrices on Riemannian manifolds, yielding robust covariance-based alignment by maximizing geodesic correlation [13].

SRM offers a probabilistic alternative to HA by aligning neural responses via maximum-likelihood estimation of a shared latent timecourse [10]. Subsequent work introduced a multi-view convolutional autoencoder (CAE + SRM), which leverages convolutional neural networks to extract richer features before alignment [23]. Matrix-Normal SRM (MN-SRM) employs Kronecker-separable covariance priors and maximum a posteriori estimation to jointly model spatial and temporal noise [24], while robust SRM (RSRM) applies sparse, deterministic optimization to disentangle shared and subject-specific components [25]. FastSRM presents an identifiable SRM variant with a dimension-reduction preprocessing step that stabilizes and accelerates shared response recovery, achieving orders-of-magnitude speed-ups without loss of accuracy [6].

Shared independent component analysis (ShICA) replaces the CCA step with independent component analysis (ICA) to learn statistically independent shared components under additive Gaussian noise, improving alignment on data with non-Gaussian artifacts [26]. However, ShICA only models shared variance. To address this, shared and individual ICA (ShIndICA) was proposed to jointly recover both shared and subject-specific sources, with provable identifiability via likelihood-based estimation [3]. Beyond ICA, the hyper hidden Markov model (Hyper-HMM) projects voxels into a latent event space and aligns temporal segments across subjects, enabling joint spatial–temporal correspondence in naturalistic fMRI paradigms [4].

Several multi-site transfer-learning approaches have been developed to harmonize task-based fMRI data across scanners and cohorts, including maximum independence domain adaptation (MIDA) [27], multi-dataset dictionary learning (MDDL) [28], and multi-dataset multi-subject (MDMS) [28], side information dependence regularization (SIDeR) [7]. The shared space transfer learning (SSTL) [1] further extends this line by extracting site-specific common features through a single-iteration multi-view optimization and mapping them into a site-independent shared space, thereby enabling scalable alignment of high-dimensional fMRI data. SSTL can incorporate the deep-kernel formulation introduced in DHA [9]—termed DeepSSTL—to further boost prediction accuracy in multi-site fMRI studies [1]. Explainability-generalizable graph neural networks (XG-GNN) is a meta-learning framework with domain-generalizable explainability regularizers that learns graph neural networks for multi-site fMRI analysis, demonstrating robust cross-center performance and interpretable subgraph discovery [8].

Self-supervised 'constructive' learning methods have recently been applied to multi-view fMRI alignment. Foundational contrastive frameworks such as SimCLR [14], bootstrap-your-own-latent (BYOL) [15], DINO [16], and its successor DINOv2 [17] learn view-invariant representations without labels. In task-based fMRI, MindEye combines contrastive encoding with diffusion priors to reconstruct viewed images while implicitly aligning subjects in the latent space [18]; MindEye2 demonstrates that a shared-subject generative model pretrained across participants can be fine-tuned with just one hour of data to decode images from fMRI [19]. More recently, MindAligner learns explicit cross-subject transformation networks for functional alignment in task-based fMRI [20]. These self-supervised approaches offer promising alternatives for multi-subject alignment by leveraging rich augmentations and momentum-based architectures. In the following, we empirically compare our proposed method to some of these approaches.

## 3 The Proposed Orthogonal Contrastive Learning (OCL)

This section introduces orthogonal contrastive learning (OCL), a novel multi-view framework for multi-subject fMRI analysis. Similar to previous functional alignment methods, OCL considers each subject's neural responses as a separate view of the same underlying data. As Figure 1 illustrated, OCL employs two neural networks with identical architectures: an online network, which actively learns representations through a contrastive objective, and a target network, whose parameters are gradually updated as a moving average of the online network's parameters. This dual-network

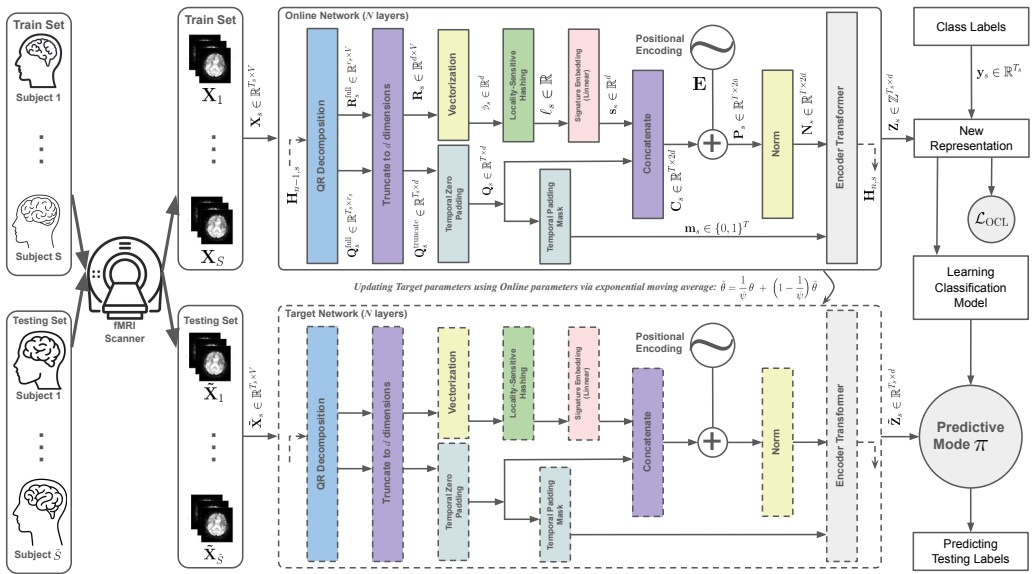

Figure 1: The Proposed Orthogonal Contrastive Learning (OCL)

setup stabilizes training and ensures consistent representations across subjects. Each OCL layer has four primary components: first, QR decomposition, which yields orthonormal feature bases to decorrelate signals and enhance the signal-to-noise ratio; second, locality-sensitive hashing (LSH), which produces compact subject-specific signatures; third, positional encoding, which integrates fMRI temporal information into spatial feature representations; finally, a transformer encoder integrates these inputs, ensuring that neural representations from the same stimulus become closely aligned, while representations of different stimuli remain distinct. In the remainder of this section, we mathematically define OCL, present a pretraining strategy to improve its performance, and discuss extending OCL through transfer learning for multi-site fMRI analysis.

We let $S$ be the number of subjects in the training set. Let $T_s, s = 1, \ldots, S$ denote the number of time points for the $s$-th subject, and let $V$ be the number of voxels in the selected region of interest (ROI), which we view as a 1D vector, even though it corresponds to a 3D volume. The preprocessed neural responses for the $s$-th subject is then defined as $\mathbf{X}_s \in \mathbb{R}^{T_s \times V}$. For simplicity, this paper assumes that each column of the neural responses is standardized during preprocessing: $\mathbf{X}_s \sim \mathcal{N}(0, \mathbf{I}), \ s = 1, \ldots, S, \mathbf{I}$ is the identity matrix. In addition, the $v$-th column in $\mathbf{X}_s$ for all subjects denotes the anatomically aligned voxel located at the same locus across fMRI images [1]. We then let $N$ defines the number of layers in the online and target networks, $f_n, n = 1, \ldots, N$ is the transformations implemented by each layer, and $\mathbf{H}_{n,s} = f_n(\mathbf{H}_{n-1,s}), n = 1, \ldots, N, s = 1, \ldots, S$ is all transformations applied in the $n$-th layer of the proposed OCL architecture on the nerual responses of $s$-th subject. Here, we consider $\mathbf{H}_{0,s} = \mathbf{X}_s$ as input layer of the networks for each subject. Further, we let $r_s = \text{rank}(\mathbf{X}_s)$ denote the rank of the neural response matrix of the $s$-th subject, and $d \leq \min_{s=1,\ldots,S}(r_s)$ be the number of components in the final representation such that $\mathbf{H}_{n,s} \in \mathbb{R}^{T_s \times d}, n = 1, \ldots, N, s = 1, \ldots, S, \mathbf{Z}_s \in \mathbb{R}^{T_s \times d} = \mathbf{H}_{N,s}$ as the final representation of neural representations for each subject that can be used for downstream classification analysis.

The first component in each OCL transformation layer is a QR decomposition. Given the neural response matrix of the $s$-th subject $\mathbf{X}_s$, we apply thin (reduced) QR decomposition [29] to factor $\mathbf{X}_s = \mathbf{Q}_s^{\text{full}} \mathbf{R}_s^{\text{full}}$, where $\mathbf{Q}_s^{\text{full}} \in \mathbb{R}^{T_s \times r_s}$ is orthonormal and $\mathbf{R}_s^{\text{full}} \in \mathbb{R}^{r_s \times V}$ is upper-triangular. We then truncate these factors by taking the first $d$ columns of $\mathbf{Q}_s^{\text{full}}$ to form $\mathbf{Q}_s^{\text{truncate}} \in \mathbb{R}^{T_s \times d}$ and the first $d$ rows of $\mathbf{R}_s^{\text{full}}$ to form $\mathbf{R}_s \in \mathbb{R}^{d \times V}$. This truncation component ensures that each OCL layer produces exactly $d$ features, where $d$ is usually the maximum number of shared features that can be extracted across all training subjects, as determined by their ranks [30]. We let $T \geq \max_{s=1,\ldots,S} T_s$ be the maximum content-window size in the OCL architecture. The temporal zero-padding component adds zero rows to each $\mathbf{Q}_s^{\text{truncate}}$ so that all orthonormal matrices share the common shape $\mathbf{Q}_s \in \mathbb{R}^{T \times d}$. Note that $T$ must be large enough to accommodate the neural responses of every subject, including those in the testing set. We also define a binary mask vector $\mathbf{m}_s \in \{0, 1\}^T$, where a value of 1 indicates an actual response time point and 0 indicates padding. We also let $d$ be

the number of nonzero elements of the upper-triangular matrix $\mathbf{R}_s$ and define $\phi_s \in \mathbb{R}^{\bar{d}} = \mathrm{vec}(\mathbf{R}_s)$ as the vectorization operator that extracts the nonzero elements of the upper-triangular matrix into a single vector, which is then used in the next step to produce subject-specific signatures.

Locality-sensitive hashing (LSH) component is a randomized, data-independent hashing scheme for approximate nearest neighbor search in high-dimensional spaces, ensuring that similar items collide with higher probability than dissimilar ones [21]. We let the parameter $p \in (0, 2]$ be the stability exponent [21, 31], $\mathbf{a} \in \mathbb{R}^{\bar{d}}$ is drawn from a $p$-stable (Gaussian) distribution, $w \in \mathbb{R}_{>0}$ be the quantization granularity of hash bins, and $b \sim \mathrm{U}[0, w]$, yielding provable locality sensitivity [21, 32]. We denote the LSH for $s$-$th$ subject as follows:

$$\ell_s \;=\; \mathrm{lsh}_{\{\mathbf{a}, b, w\}}(\phi_s) \;=\; \left\lfloor \frac{\langle \mathbf{a}, \, \phi_s \rangle \;+\; b}{w} \right\rfloor, \tag{1}$$

where $\lfloor \, \rfloor$ is the floor function. Note that LSH is identical for all neural responses of the $s$-$th$ subject and needs only be computed once for all time points belonging to that subject in each training iteration. We then use a linear multilayer perceptron (MLP) that accepts the scalar $\ell_s \in \mathbb{R}$ as input and produces the vector $\mathbf{s}_s \in \mathbb{R}^d$ as the subject-specific signature (embedding) for the $s$-$th$ subject.

**Lemma 1.** *Let each neural responses matrix admit a reduced QR factorization $\mathbf{X}_s = \mathbf{Q}_s \mathbf{R}_s$ with $\mathbf{Q}_s^\top \mathbf{Q}_s = \mathbf{I}$. Define $\phi_s = \mathrm{vec}(\mathbf{R}_s) = \mathrm{vec}(\mathbf{Q}_s^\top \mathbf{X}_s)$. If*

$$\big\| \phi_1 - \phi_2 \big\| < \big\| \phi_1 - \phi_3 \big\| \rightarrow \Pr\big[ \ell_1 = \ell_2 \big] \;>\; \Pr\big[ \ell_1 = \ell_3 \big].$$

Please refer to the supplementary material for the proof.

We let $\mathbf{C}_s \in \mathbb{R}^{T \times 2d}$ be the output of the concatenation component, which combines each row of the orthonormal matrix $\mathbf{Q}_s \in \mathbb{R}^{T \times d}$ with the subject-specific signature $\mathbf{s}_s \in \mathbb{R}^d$. We then apply a sinusoidal positional encoding to embed temporal context alongside the spatial and signature features. Concretely, we construct a positional encoding matrix

$$\mathbf{E} \in \mathbb{R}^{T \times 2d} = [e_{1,0}, \ldots, e_{T,2d}], \quad e_{t,2i} = \sin\!\Big( \frac{t}{T^{2i/(2d)}} \Big), \quad e_{t,2i+1} = \cos\!\Big( \frac{t}{T^{2i/(2d)}} \Big),$$

for $t = 1, \ldots, T$ and $i = 0, \ldots, d-1$. Adding this to the concatenated features yields the component $\mathbf{P}_s \in \mathbb{R}^{T \times 2d} = \mathbf{C}_s + \mathbf{E}$, which now encodes both spatial patterns and their temporal positions. Next, OCL applies a normalization component to stabilize and scale each time-step embedding, *i.e.*, given the positional-encoded features $\mathbf{P}_s \in \mathbb{R}^{T \times 2d}$, we compute the normalization $\mathbf{N}_s \in \mathbb{R}^{T \times 2d} = \mathrm{Norm}(\mathbf{P}_s)$. Finally, these normalized embeddings are passed, together with the temporal padding mask $\mathbf{m}_s$, into a standard Transformer encoder [33] to produce corresponding layer output $\mathbf{H}_{n,s}$.

We let $\mathbf{y}_s = [y_{s,1}, \ldots, y_{s,T_s}]^\top \in \mathbb{R}^{T_s}$ denote the class labels for the $s$-th subject in the training set. For the $s$-th subject, let $\mathbf{Z}_s = [z_{s,1}, \ldots, z_{s,T_s}]^\top \in \mathbb{R}^{T_s \times d}$ be the output of final layer of the OCL online network. We define the contrastive loss $\mathcal{L}_{\mathrm{OCL}}(\mathbf{Z}_s, \mathbf{y}_s)$ with temperature $\tau$, margin $\mu$, and between-class weight $\lambda$ for $s$-th subject as

$$\mathcal{L}_{\mathrm{OCL}\{\tau, \mu, \lambda\}}(\mathbf{Z}_s, \mathbf{y}_s) = -\frac{1}{T_s} \sum_{i=1}^{T_s} \log \frac{\displaystyle\sum_{\substack{j=1 \\ j \neq i, \, y_{s,j} = y_{s,i}}}^{T_s} \exp\big( \langle z_{s,i}, \, z_{s,j} \rangle / \tau \big)}{\displaystyle\sum_{k=1}^{T_s} \exp\big( \langle z_{s,i}, \, z_{s,k} \rangle / \tau \big)} \tag{2}$$

$$+ \lambda \frac{1}{T_s^2} \sum_{i=1}^{T_s} \sum_{\substack{j=1 \\ y_{s,j} \neq y_{s,i}}}^{T_s} \log\Big( 1 + \exp\big( \langle z_{s,i}, \, z_{s,j} \rangle / \tau \; - \; \mu \big) \Big).$$

Let $\theta$ denotes all learnable parameters of the online encoder and $\tilde{\theta}$ those of the target encoder. In each training iteration for subject $s$, we first update the online parameters by one step of gradient descent on the subject's contrastive loss: $\theta \leftarrow \theta - \eta \nabla_\theta \mathcal{L}_{\mathrm{OCL}}(\mathbf{Z}_s, \mathbf{y}_s)$, $\eta$ is the learning rate. We update the online network using all subjects in the training set during each iteration. Once the online network has processed every subject's data in each iteration, we update the target network parameters using an exponential moving average (EMA) of the online parameters as follows [15]:

$$\tilde{\theta} = \frac{1}{\psi} \theta \;+\; \Big( 1 - \frac{1}{\psi} \Big) \tilde{\theta}, \tag{3}$$

Table 1: The fMRI datasets

| ID | Title | Type | $S$ | $|\mathbf{y}|$ | $T_s$ | Site(#) |
|---|---|---|---|---|---|---|
| A* | Stop signal (DS007) [34] | Decision | 20 | 4 | 472 | B (3) |
| B | Conditional stop signal (DS008) [35] | Decision | 13 | 4 | 317 | A (1) |
| CMU | Meanings of Nouns [36] | Semantic | 9 | 12 | 402 | |
| C | Simon task (DS101) (unpublished [7]) | Simon | 21 | 2 | 302 | D (1) |
| D | Flanker task (DS102) [37] | Flanker | 26 | 2 | 292 | C (1) |
| DS232 | Face-coding models with individual-face [38] | Visual | 10 | 4 | 760 | |
| E | Integration of sweet taste: Study 1 (DS229) [39] | Flavour | 15 | 6 | 580 | F (1) |
| F | Integration of sweet taste: Study 3 (DS231) [39] | Flavour | 9 | 6 | 650 | E (1) |
| Forrest | Forrest Gump movie [40] | Visual | 20 | 10 | 451 | |
| Raiders | Raiders movie [5, 10] | Visual | 10 | 7 | 924 | |

$S$ is the number of subjects; $|\mathbf{y}|$ is the number of stimulus categories; $T_s$ is the number of time points per subject; *Site* lists the other datasets whose neural responses can be transferred to this dataset. # represents the number of sites in the corresponding dataset. * this dataset is partitioned into three independent 'sites'—pseudo-word naming (A1), letter naming (A2), and manual response (A3) [1]

where $\psi$ is the number of total iterations. In the training phase, OCL learns a shared representation space in which neural recordings from all training subjects are aligned. We then train a classifier (denoted by $\pi$ in Figure 1) on these new representations. In the testing phase, we use the trained target network to map the test data into the same representation space and then apply the classifier to predict cognitive states. We provide the pseudocode for the proposed OCL algorithm in the supplementary material.

## 3.1 General Pretrained Orthogonal Contrastive Model

To bootstrap OCL for real task-based fMRI, we first pretrain the dual-encoder entirely on synthetic data that mimics the statistical structure of neural timecourses. Concretely, given $k$ class categories, we generate a corpus of random base matrices $\mathbf{M} \in \mathbb{R}^{T \times V}$, where each group of $\frac{T}{k}$ rows is drawn *i.i.d.* from one of $k$ distinct Gaussian distributions with randomly initialized means and variances that differ across distributions. For each base matrix $\mathbf{M}$, we then create $S$ distinct 'views' by applying $S$ random orthonormal rotations: $\mathbf{X}_s = \mathbf{M}\,\mathbf{U}_s, \quad \mathbf{U}_s^\top \mathbf{U}_s = I, \quad s = 1, \ldots, S$. Since each row of $\mathbf{M}$ is sampled from one of the $k$ distributions, we assign a corresponding class label $y = k$ to that row, and this label is preserved across all rotated views $\{\mathbf{X}_s\}_{s=1}^S$. Each view $\mathbf{X}_s$ is passed through the OCL layers to produce embeddings $\mathbf{Z}_s$, *i.e.*, the contrastive objective pulls together embeddings of the same label across different rotations and pushes apart embeddings of different labels. After pretraining, we transfer the *target* encoder's parameters $\tilde{\theta}$ to initialize the downstream OCL model on the real fMRI data. This EMA-smoothed encoder has already learned to factor out arbitrary orthogonal transforms and subject-specific variability, providing a strong, generalizable starting point for aligning real neural data with minimal additional tuning.

**Lemma 2.** *Let each synthetic view be generated by an orthonormal rotation of the base data,* $\mathbf{X}_s = \mathbf{M}\,\mathbf{U}_s, \quad \mathbf{U}_s^\top \mathbf{U}_s = \mathbf{I}$. *Write the corresponding QR factor* $\mathbf{R}_s$ *and its flattened vector* $\phi_s = \mathrm{vec}(R_s)$. *Then for any three views* $\mathbf{X}_1, \mathbf{X}_2, \mathbf{X}_3$ *generated from* $\mathbf{M}$,

$$\Pr\big[\ell_1 = \ell_2\big] \;=\; \Pr\big[\ell_1 = \ell_3\big].$$

*In other words, the collision probability of the LSH hash is identical across all random rotations generated from* $\mathbf{M}$.

Please see the supplementary material for the proof.

## 3.2 Transfer Learning via Orthogonal Contrastive Embeddings

To extend OCL to multi-site fMRI studies, suppose we have $B$ training sites, each providing data $\{\mathbf{X}_s^{(b)}, \mathbf{y}_s^{(b)}\}_{s=1}^{S_b}$ for site $b = 1, \ldots, B$. We train an independent OCL instance on each site, yielding target-encoder parameters $\tilde{\theta}_b, \; b = 1, \ldots, B$. These *site-specific* encoders capture local scanner and population idiosyncrasies while maintaining the shared contrastive objective. To initialize OCL on a multi-site setup (with no extra fine-tuning step), we aggregate the $B$ learned targets via

$$\tilde{\theta}_{\text{sites}} \;=\; \frac{1}{B} \sum_{b=1}^{B} \tilde{\theta}_b, \tag{4}$$

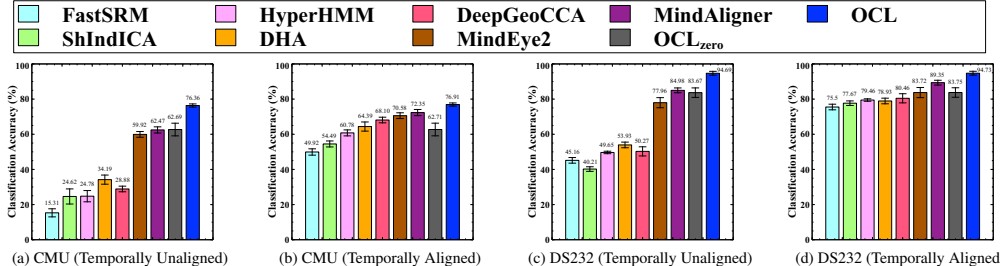

Figure 2: Classification analysis on Temporally Aligned versus Temporally Unaligned Data. Plotted are mean accuracies and error bars are $\pm 1$ standard deviation.

thereby blending all site-specific knowledge into a single robust prior. We then freeze $\tilde{\theta}_{\text{sites}}$ and apply it to both the training-site and testing-site data, projecting each into the same shared feature space defined by the aggregated targets—this alignment boosts the accuracy of our downstream classifiers.

Please note that this multi-site adaptation scheme also enables an active learning loop in the real world applications. At each iteration, we evaluate the contrastive loss $\mathcal{L}_{\text{OCL}}^{(\bar{b})}$ on the predicted time points from the testing site $\bar{b}$ and select those with highest uncertainty (e.g., largest margin-based loss) for expert manual labeling. The newly annotated samples are then incorporated into the training set, the online encoder parameters are updated accordingly, and the target encoder is refined via EMA. By focusing annotation effort on the most informative temporal segments, this closed-loop procedure maximizes performance gains in low-data or high-cost labeling scenarios. Although active learning offers a promising avenue for extending OCL, it is beyond the scope of this study; all reported results were obtained without active-learning or expert-annotated data.

## 4 Experiments

Table 1 summarizes the 10 datasets used in our empirical evaluation. Datasets CMU and DS232 consist of simple cognitive task-prediction paradigms, whereas Forrest and Raiders datasets involve naturalistic movie-watching stimuli in single-site fMRI studies. We also include 6 homogeneous task-based fMRI datasets (A–F) suitable for multi-site analysis. All datasets are publicly available (via OpenNEURO [4], except CMU [5]) and were preprocessed with our GUI-based toolbox called easy fMRI [6] and FSL 6.0.15 [7], including spatial normalization, smoothing, anatomical alignment; for those alignment techniques that require it, temporal realignment was also applied (see Section 4.1). Each scan was registered to the MNI152 T1-weighted template [1] at a $4mm$ isotropic resolution, and a whole-brain ROI was defined for all analyses, yielding $V = 19{,}742$ voxels per volume. Data were standardized during preprocessing, without loss of generality.

We benchmark OCL against 7 single-site fMRI analysis methods: FastSRM and HyperHMM as baselines; ShIndICA as a non-CCA method; DHA and DeepGeoCCA as deep multi-view learning approaches; and MindEye2 and MindAligner as self-supervised constructive learning approaches. For multi-site evaluation, we compare OCL to 5 existing techniques: SSTL as a baseline; DeepSSTL and XG-GNN as deep multi-site learning approaches; and MindEye2 and MindAligner as self-supervised constructive methods. Crucially, each site's data are strictly partitioned so that no neural responses from a given site appear in both the training and testing sets. All experiments were run on two PCs with the specifications listed in the Footnote [8]. Our proposed OCL algorithm is available on GitHub [9]. Like the previous studies [1, 9, 10], we employ a $\nu$-support vector machine ($\nu$-SVM) [41] for all classification experiments. We use a leave-one-subject-out nested cross-validation: in each outer fold, one subject is held out for testing; within each, another subject serves as validation (inner fold), and the rest form the training set. Hyperparameters for alignment and $\nu$-SVM (e.g., RBF kernel scale, $\nu$) are selected via grid search on validation accuracy, and the best testing accuracy is reported for each technique.

---

[5] Available at `https://www.cs.cmu.edu/afs/cs.cmu.edu/project/theo-81/www/`

[6] Available at `https://easyfmri.learningbymachine.com/`

[7] Available at `https://fsl.fmrib.ox.ac.uk/fsl`

[8] OS: Fedora 42, Python: 3.11.9, PyTorch: 2.6, CUDA: 12.6; Connection: $2\times40\text{GbE}$ CX314A Mellanox (PC1) CPU: AMD EPYC $7551P$ (64 cores), RAM: 256G GPU:$2\times$NVIDIA 4060Ti 16G; (PC2) CPU: AMD Threadripper $2990WX$ (64 cores), RAM: 128G, GPU:$2\times$NVIDIA 4060Ti 16G.

[9] OCL code repository: `https://github.com/myousefnezhad/ocl`

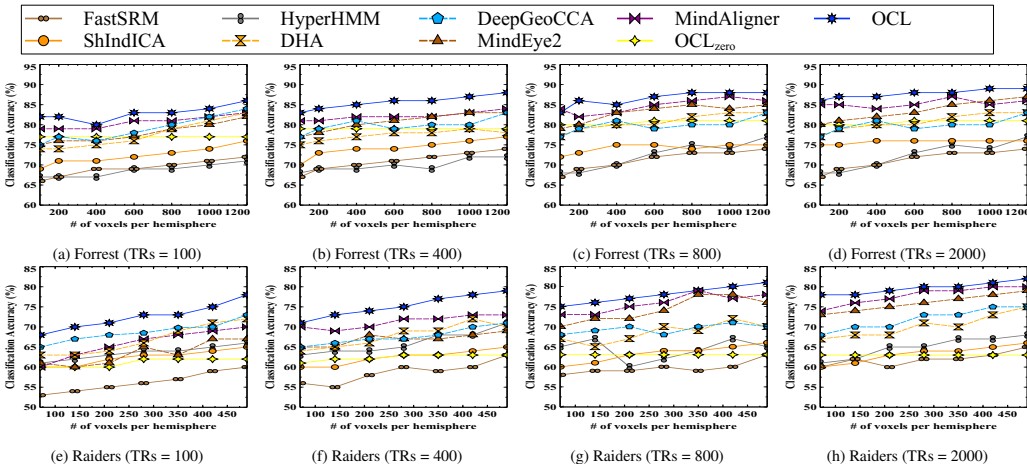

Figure 3: Classification analysis on movie stimuli. Potted are mean accuracies.

For OCL, we first initialize a model ('OCL$_{\text{zero}}$') by pretraining on two million ($4 \times 500{,}000$) synthetic matrices $\mathbf{M} \in \mathbb{R}^{T \times V}$ for $k \in \{5, 10, 20, 50\}$ categories with $T = 2000$ time points and $V = 19742$ voxels, each subjected to $S = 360$ random rotations. We set the embedding dimension to $d = 256$, employ Encoding Transformer with 16 attention heads, and use $N = 32$ network layers. These hyperparameters were chosen to balance representational capacity with available computational resources listed in the Footnote [8]. We then initialize each OCL instance with the pretrained target encoder $\tilde{\theta}$, and fine-tune both online and target networks on the real fMRI data. We train (and pretrain) OCL for up to $\psi = 1000$ iterations with automatic early stopping based on validation loss using Adam optimizer [42]. At each iteration, we form a batch from all time points of a single subject—treating each time point as an independent sample—and randomly shuffle their order. This permutation prevents the network from overfitting to a fixed temporal sequence and encourages robustness to varied response orderings. Because every fine-tuned OCL model is initialized from OCL$_{\text{zero}}$, we employ the same OCL network architecture across all experiments in this paper. OCL consistently produces a feature space of dimension $d = 256$. For all competing methods, we evaluate two latent space sizes—one fixed at $d = 256$ and a second chosen via grid search—and report the configuration that yields the highest classification accuracy. Other self-supervised approaches are similarly initialized with their published pretrained weights [19, 20]. All remaining hyperparameters for both the alignment and classification models are optimized using grid search. We perform grid search over the key OCL hyperparameters — temperature $\tau \in \{0.01, 0.1, 0.5, 0.9, 0.99\}$, margin $\mu \in \{0.1, 0.2, 0.5, 0.8, 0.9\}$, between-class weight $\lambda \in \{0.1, 0.2, 0.3, 0.4, 0.5\}$, learning rate $\eta \in \{0.1, 0.2, 0.3, 0.4\}$, and quantization granularity $w \in \{0.9, 1.0, 1.1, 1.2\}$ — and select the combination that maximizes performance on the validation set.

## 4.1 Simple Cognitive Task Classification: Temporally Aligned *vs.* Unaligned Data

This section evaluates OCL on two simple cognitive-task datasets (CMU and DS232), in which subjects performed Semantic, and Visual assessments during fMRI scanning. Unlike most functional alignment methods—which assume temporal synchronization (*i.e.*, each time point $t$ corresponds to the same stimulus across subjects)—OCL can handle varying time-series lengths and arbitrary time-point ordering without explicit temporal preprocessing. We therefore compare OCL and several alignment techniques both on the raw, unaligned data and after applying their required temporal alignment. To robustly tune and assess performance, we employ a nested leave-one-subject-out procedure: in each outer fold one subject is held out for testing, while in each inner fold a different subject is held out for validation and hyperparameter selection. Figure 2 shows that traditional alignment methods suffer significant testing accuracy degradation on unaligned data because their shared-space templates misalign stimuli across rows; in contrast, self-supervised constructive approaches (MindEye2, MindAligner, and OCL) learn flexible mappings rather than fixed templates, yielding stable shared representations. OCL in particular achieves the highest accuracy, likely due to (1) its orthogonal decomposition of independent versus subject-specific features and (2) the pretrained 'OCL$_{\text{zero}}$' encoder's ability to generalize arbitrary rotations to novel neural patterns. Each of the 4 plots in Figure 2 is comparing OCL with 7 different methods $\chi = \{$FastSRM, ShIndICA,

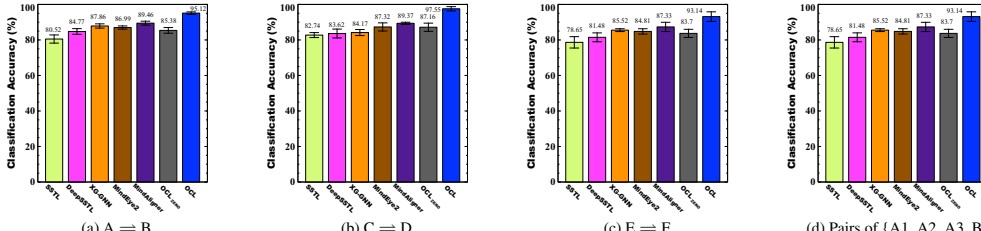

| (a) A ⇌ B | (b) C ⇌ D | (c) E ⇌ F | (d) Pairs of {A1, A2, A3, B} |

Figure 4: Multi-site Classification Analysis. Plotted are mean accuracies and error bars are ±1 standard deviation.

HyperHMM, DHA, DeepGeoCCA, MindEye2, MindAligner} for a total of $4 \times 7 = 28$ comparisons — where the 2-sided t-test found $\rho < 0.05$ in all 28 cases.

## 4.2 Classification Analysis on Movie Stimuli

This section evaluates functional alignment techniques on the two movie-watching fMRI datasets (Forrest and Raiders) listed in Table 1, using the same nested leave-one-subject-out scheme as in Section 4.1. Following the procedure of [5], we first rank voxels within the predefined ROI by their task-evoked activation strength, as in [9, 10, 30]. We then assess each alignment method across multiple spatial resolutions by selecting the top $\{100, 200, 400, 600, 800, 1000, 1200\}$ voxels for Forrest and $\{70, 140, 210, 280, 350, 420, 490\}$ voxels for Raiders. To test temporal coverage, we further repeat all experiments using the first $\{100, 400, 800, 2000\}$ Time of Repetitions (TRs) of each scan. Note that in OCL, voxels outside the ROI are zeroed out in the input layer, and the subsequent QR decomposition automatically ignores these zeros, preventing them from influencing the learned representations. Figure 3 presents classification accuracy as a function of voxel count and number of time points, comparing our proposed OCL to 7 alignment approaches $\chi = \{$FastSRM, ShIndICA, HyperHMM, DHA, DeepGeoCCA, MindEye2, MindAligner$\}$ for a total of $8 \times 7 \times 7 = 392$ pairwise evaluations. Self-supervised methods initialized with pretrained models typically outperform traditional functional-alignment techniques. In every evaluation, OCL surpasses all comparators—an advantage we attribute to its orthogonal feature decomposition and contrastive alignment. A two-sided $t$-test confirms that the accuracy differences are significant ($\rho < 0.05$) in all 392 comparisons.

## 4.3 Multi-Site Classification Analysis

This section presents the results of multi-site fMRI analyses using datasets A–F listed in Table 1. For each pair of sites $(A, B)$, we conducted a two-sided cross-site evaluation: in the forward direction $A \to B$, we trained and validated alignment and classification models using site A and evaluated them on site B; in the reverse direction $B \to A$, we reversed the training and testing roles. The final accuracy is computed as the mean of both directions. This bidirectional setup is denoted as $A \rightleftharpoons B$. Figures 4 (a–c) summarize the classification accuracies across multiple cross-site pairs. In addition, Figure 4 (d) presents the mean accuracy of supplementary experiments based on all possible two-versus-two train/test splits derived from the set {A1, A2, A3, B}, yielding six distinct comparisons[10]—for example, training on {A1, A2} and testing on {A3, B}, and vice versa. As shown, the baseline method SSTL, which relies on linear transformations, consistently underperforms. In contrast, DeepSSTL improves accuracy by utilizing a MLP based deep kernel for alignment. Constructive learning approaches further enhance accuracy by leveraging pretrained models that better generalize across domains. The proposed OCL framework achieves the highest classification performance across all site-pairs. This improvement appears to stem from two key design elements: (1) the use of a pretrained multi-representational alignment module that provides a robust initial feature space, and (2) a specialized architecture that enforces cross-site shared information via a contrastive loss. Each of the 4 subplots in Figure 4 compares OCL against a competing method $\chi$, for each of five baselines $\chi \in \{$SSTL, DeepSSTL, XG-GNN, MindEye2, MindAligner$\}$, resulting in a total of $4 \times 5 = 20$ comparisons. In all cases, a two-sided paired $t$-test yielded statistically significant differences ($\rho < 0.05$), confirming the robustness of OCL in cross-site generalization.

---

[10]Pairs: {A1, A2}, {A1, A3}, {A1, B}, {A2, A3}, {A2, B}, {A3, B}

# 5 Conclusion

This paper has introduced orthogonal contrastive learning (OCL), a unified framework that addresses task-based fMRI's key challenges: low signal-to-noise ratio, high dimensionality, and variable time-series lengths. OCL aligns neural responses across subjects and sites without explicit temporal preprocessing. OCL employs a dual-encoder design: an online network and a target network whose weights track the online network via exponential moving average to stabilize learning. Each OCL network layer combines QR decomposition for orthogonal feature extraction, locality-sensitive hashing (LSH) to produce compact subject-specific signatures, positional encoding to embed temporal structure alongside spatial features, and a transformer encoder to generate discriminative, stimulus-aligned embeddings, trained with a contrastive loss that pulls together same-stimulus responses and pushes apart different-stimulus responses. We further enhance OCL with an unsupervised pretraining stage on fMRI-like synthetic data and demonstrate a transfer-learning workflow for multi-site studies. Across extensive experiments on multi-subject and multi-site fMRI benchmarks, OCL consistently outperforms state-of-the-art alignment and analysis methods in both representation quality and downstream classification accuracy.

In the future, OCL has the potential to be applied across a variety of task-based fMRI studies, such as reconstructing visual stimuli or movies from human brain activity, as well as extended to other neuroimaging modalities including resting-state fMRI, MEG, and EEG. For resting-state fMRI, pseudo-labels can be generated by applying sliding-window functional connectivity or by clustering temporal windows based on correlation patterns; OCL can then maximize contrastive agreement across these pseudo-classes, effectively aligning subjects in the absence of explicit tasks. Similarly, for MEG and EEG data, the decomposition can operate on sensor- or source-space time series, while LSH can hash spectral or time–frequency representations. Positional encoding further preserves temporal ordering, even when sampling rates vary across modalities. Moreover, OCL can be scaled to larger voxel spaces and multi-site datasets, paving the way toward a foundation model for fMRI analysis—although such scaling would require substantial GPU cluster resources.

# 6 Broader Impacts

Orthogonal contrastive learning (OCL) enables large-scale pooling of multi-site fMRI datasets, substantially improving statistical power and reproducibility by harmonizing site-specific biases and reducing variance in group-level inferences. By aligning subject-specific neural signatures into a shared space, OCL facilitates the discovery of robust biomarkers for neurological and psychiatric disorders, advancing precision psychiatry and personalized medicine. Eliminating the need for explicit temporal preprocessing and uniform time-series lengths, OCL lowers technical barriers to integrating diverse datasets, supporting open-science platforms and accelerating collaborative research. Finally, by democratizing access to high-quality, reproducible fMRI representations and mitigating batch effects, OCL promotes ethical, transparent AI-driven neuroscience, fostering cross-disciplinary innovation and responsible research practices.

# 7 Limitations

OCL's multi-module architecture enables strong alignment but comes with important caveats: (i) Computation: its dual-network and Transformer components demand substantially greater computation power than traditional alignment methods, limiting scalability to very high-dimensional feature spaces; (ii) Domain shift: performance may degrade under extreme inter-site or inter-subject domain shifts beyond those in our benchmarks; (iii) Data requirements: its ability to generalize robustly hinges on access to large, well-annotated multi-site datasets, constraining usefulness in small-sample or weakly labeled settings; and (iv) Interpretability: deep learning in neuroimaging is not easily interpretable, and while we provide initial insights via QR-basis spatial projections and attention maps, comprehensive interpretability is outside the scope of this work.

## Acknowledgments

I would like to express my sincere gratitude to Stephen P. Kaiser of the National Bank of Canada for his generous support in the publication of this research study.

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
