# OpenReview forum: "Orthogonal Contrastive Learning for Multi-Representation fMRI Analysis"
_NeurIPS.cc/2025/Conference — NeurIPS 2025 poster_

### Official Review · Reviewer_xAGi · 2025-06-30

**Clarity:** 2
**Significance:** 2
**Originality:** 3
**Rating:** 4
**Confidence:** 4

**Summary:**

This paper propose a method called orthogonal contrastive learning (OCL) to align fMRI response across multiple subjects without the need for aligned temporal data. The method consists of QR decomposition for orthogonal feature extraction, locality-sensitive hashing to produce subject specific codes, positional encoding for embedding temporal information, and a transformer encoder to generate the embeddings.  The paper also proposes an initialization method using fMRI-like synthetic data and an approach for transfer learning for multi-site studies. Experiments on multiple datasets shows OCL outperformed multiple baselines in classification accuracy of fMRI stimuli.

**Questions:**

1. How do the different model design components contribute to the overall performance of the model?

2. What are the ROIs used in the experiments? Is it whole brain? How are they defined/decided? Are the same ROIs/inputs used for each of the baseline methods?

3. Does the performance of OCL vary greatly with changing hyper parameter values?

4. I am somewhat confused about the transfer learning section. Did the authors do the extra active learning loop annotating testing site data with greatest uncertainty, as mentioned at the bottom of p.6? Do any other baselines leverage this added information? If not the comparison does not seem quite fair.

5. Also, I'm confused about the multi-site experiment setup. It seems that only 1 site was used for training, then transferred to one for testing, and multiple sites are thus not used for training. Is this correct? Then it is not testing the multi-site approach in 3.2.

**Ethical Concerns:**

["NO or VERY MINOR ethics concerns only"]

**Final Justification:**

Please see longer response below - the authors helped point out missed ablation experiments and provided many clarifications that addressed most of my questions.

**Limitations:**

Checklist claims authors discuss limitations in conclusions/broader impacts, but I don't see such discussion.

**Quality:**

3

**Strengths And Weaknesses:**

Strengths:
1. The proposed approach does not require preprocessing of the fMRI data such that sequences must be aligned or of the same length. This aids in usability of the model and ease in combining different datasets.
2. The experiments were performed using 10 datasets (9 open), classifying different types of fMRI stimuli and using fMRI data obtained from multiple different sites.
3. The experiments involved comparison against up to 7 baselines, with some statistical analysis of results.

Weaknesses:
1. Questions about model design:
While the proposed approach does not require aligned sequences, I have concerns about the removal of considering the spatiotemporal data as a whole. While temporal encoding is included, this does not include information about the temporal stimulus information that occurs beforehand (as far as I can tell). The neural response likely depends on what information was presented before. Since the analysis is on a time-point basis, one would lose this important information.
2. Lack of ablation studies / understanding of contribution of model components:
The proposed method involves many different components. However, the experiments only assess the model as a whole, while the authors conclude the improvement is due to different factors, but there is no direct evidence/evaluation of this. The lack of ablation studies / assessment of contribution of different model components makes it so we cannot understand the advantage of different design choices, which may or may not be effective.
3. Sensitivity to hyperparameters:
The experiments do not present any sensitivity analysis of hyperparameters. Furthermore, it is clear that the many hyperparameters of OCL were tuned, but it is not clear whether the competing methods had as much care in hyper parameter tuning, and therefore there are concerns whether all models were optimized to their best.

---

> ### Author Rebuttal · Authors · 2025-07-31
>
> Dear Reviewer,
>
> Thank you for your thoughtful critique and constructive suggestions. We address each point below, citing relevant sections of our manuscript and outlining planned enhancements.
>
> 1. Ablation Studies & Component Contributions
>
> In Supplementary Material Section D, we perform ablation experiments on the CMU and DS232 datasets by systematically removing or substituting each OCL component (QR decomposition, LSH, positional encoding, and Transformer). Across all variants, we observe consistent declines in both alignment consistency and downstream classification performance relative to the full OCL model, confirming that each module contributes meaningfully to our framework.
>
> 2. Regions of Interest (ROI)
>
> As detailed in Section 3.1 and evaluated in Section 4, we employ a whole-brain ROI defined on the MNI152 T1 template (4 mm isotropic resolution), resulting in V = 19,742 voxels per volume. This single mask is used consistently during both pretraining and fine-tuning to ensure identical input dimensions across all subjects, sites, and datasets. We will include a comprehensive description of the ROI selection process in the final manuscript to enhance reproducibility.
>
> 3. Hyperparameter Sensitivity & Fair Tuning
>
> We evaluated OCL and all baseline methods using nested leave-one-subject-out cross-validation with comprehensive grid searches over key hyperparameters (e.g., learning rate, temperature, margin). Across these experiments, OCL’s classification accuracy varied by less than 5% in nearly all configurations, demonstrating stability and robustness to hyperparameter choices. Identical tuning protocols and search spaces were applied to all methods to ensure fair and consistent comparisons. We will provide further details in the final manuscript to enhance reproducibility.
>
> 4. Transfer Learning & Active Learning Loop
>
> Section 3.2 (bottom of p. 6) describes an optional active-learning loop: after aggregating multi-site encoder parameters, we identify high-uncertainty TRs for expert annotation and iteratively refine the target encoder. To ensure fair comparison, our current experiments did not employ this loop, and no expert feedback was used. While integrating active learning is a promising direction for future extensions of OCL, it falls outside the scope of this work. We will explicitly clarify in the final manuscript that all reported results are obtained without any active-learning or additional expert annotations.
>
> 5. Multi-Site Experimental Setup
>
> We apologize for any confusion regarding our site definitions. In our primary multi-site evaluation, dataset DS007 is partitioned into three independent ‘sites’—pseudo‑word naming (A1), letter naming (A2), and manual response (A3)—while dataset DS008 (B) serves as a single held-out site, following the experimental design of SSTL (NeurIPS 2020). We first train on {A1 + A2 + A3} (B=3 in Eq. 4) and test on B, then reverse by training on B and testing on each of A1, A2, and A3, averaging these bidirectional accuracies.  To further support the effectiveness of our proposed OCL framework, we report additional results based on an exhaustive set of two-versus-two train/test splits formed from the set {A1, A2, A3, B}, resulting in six unique comparisons (e.g., training on {A1, A2} and testing on {A3, B}, and vice versa). In these experiments, B = 2 in Eq. (4). Across all splits, OCL achieved a mean accuracy of %98.23 ± 0.64, significantly outperforming the second-best baseline, MindAligner (%91.63 ± 1.71). For the remaining datasets (C–F), each comprising only two scanning sites, we perform a similar train/test swap procedure (B = 1 in Eq. 4), which still qualifies as a multi-site analysis. We will provide a detailed explanation of these protocols along with comprehensive results in the final manuscript.
>
> Thank you once again for your thoughtful and constructive feedback. We hope that the revisions satisfactorily address your concerns and effectively underscore the practical value and scientific contributions of OCL. We sincerely appreciate your consideration and respectfully request your support for the publication of our manuscript.
>
>
> Sincerely,
>
> The Author(s)

---

> ### Comment · Reviewer_xAGi · 2025-08-08
> **Response to rebuttal**
>
> I thank the authors for the detailed rebuttal, which clarifies many of my concerns, including the missed ablation studies, the hyper parameter sensitivity/tuning analysis (which should be added in a revision), and the transfer/active learning discussion (which needs to be clarified in a revision). My concern regarding motivation of model design and lack of consideration of the spatiotemporal information remains, but given the added information in the rebuttal and the promise of including the clarifications in a final revision, I will increase my score from 3 to 4.

---

### Official Review · Reviewer_xXsd · 2025-07-01

**Clarity:** 2
**Significance:** 3
**Originality:** 3
**Rating:** 4
**Confidence:** 4

**Summary:**

This paper proposes Orthogonal Contrastive Learning (OCL), a novel multi-representation framework designed for multi-subject and multi-site task-based fMRI analysis. The method integrates several components—QR decomposition, locality-sensitive hashing (LSH), positional encoding, and a transformer encoder—within a dual-encoder architecture (online and target networks). The model is trained using a contrastive loss that aligns same-stimulus fMRI responses across subjects while preserving distinctiveness across different stimuli. A pretraining phase on synthetic fMRI-like data provides a robust initialization, followed by fine-tuning on real data. Extensive experiments demonstrate that OCL outperforms various state-of-the-art functional alignment and contrastive learning methods in both representation quality and downstream classification performance.

**Questions:**

1. How critical are QR decomposition and LSH individually? Could the authors provide ablation studies isolating these components?
2. This method involves large Transformer networks and QR decomposition, both of which require training. Given the inherently noisy nature of fMRI data, it raises concerns about whether such a complex architecture is well-suited to this domain. Additionally, these components may introduce computational bottlenecks. How does the training time and memory usage compare to other approaches that leverage pretrained models without the need for such heavy computation?
3. The positional encoding used in this paper has already been widely applied in the neuroimaging field. Could the authors clarify the specific purpose of using positional encoding in their framework? Is there any methodological innovation in how it is applied here, or is it directly borrowed from existing practices without adaptation?

**Ethical Concerns:**

["NO or VERY MINOR ethics concerns only"]

**Final Justification:**

After considering the authors’ rebuttal, I have decided to raise my score from 3 to 4. The rebuttal effectively addressed most of my major concerns. Although some omissions mentioned in my comments reduce the perceived completeness and generality of the work, I believe the core contribution is technically solid and relevant to the community. I look forward to the authors’ final revised and polished manuscript.

**Limitations:**

The paper does not include a dedicated Limitations section, and the limitations mentioned in the main text are relatively sparse. It is recommended that the authors provide further discussion. For example, considering that the proposed method may potentially be applied to sensitive tasks such as cognitive state prediction, mental health diagnosis, or multi-subject generation, the authors are encouraged to briefly discuss the potential societal impact of such applications, including risks of misuse, and offer suggestions for mitigation. In addition, the authors should discuss the scalability boundaries of their approach and analyze the trade-offs between accuracy and computational efficiency, especially given the method's high computational cost.

**Paper Formatting Concerns:**

None formatting issue

**Quality:**

2

**Strengths And Weaknesses:**

Strengths

1. The paper is clearly written and well-structured, especially the explanation of the OCL architecture.
2. The mathematical principles proposed in this paper are all rigorously proven and thoroughly presented, reflecting the authors' strong mathematical foundation.
3. The paper addresses a well-motivated and difficult problem in multi-subject fMRI analysis and demonstrates tangible improvements over strong baselines in challenging benchmarks.

Weaknesses

1. The paper takes a highly mathematical approach in selecting its methodological framework, incorporating techniques such as QR decomposition and locality-sensitive hashing (LSH). Although the authors provide proofs for the alignment procedures and technical details, what is more crucial is validating the effectiveness of these methods in multi-subject fMRI tasks. The current work lacks theoretical justification for their domain-specific applicability—for example, whether similar approaches have been previously applied in this field. If the proposed method is indeed pioneering, more detailed ablation studies should be conducted to verify its effectiveness, rather than relying solely on mathematical derivations.
2. The paper explores only classification tasks in the experimental section, which is a rather basic evaluation for fMRI decoding. Given that many recent works—especially those using mainstream models like CLIP—have already achieved high classification accuracy, the current evaluation setting appears limited. The authors are encouraged to explore additional tasks such as retrieval and reconstruction to better demonstrate the generality and broader applicability of their proposed method.
3. The balance between theory and practice in this paper is not well maintained— the theoretical part is disproportionately emphasized, while the experimental section is relatively limited. Moreover, the presentation of results is overly simplistic, relying solely on bar charts without providing the underlying numerical data in tables or other forms. This limits the clarity and reproducibility of the experimental findings.

---

> ### Author Rebuttal · Authors · 2025-07-31
>
> Dear Reviewer,
>
> Thank you for your thorough evaluation and constructive suggestions. Below, we address each of your concerns in detail.
>
> 1. Criticality of QR Decomposition and LSH
>
> In Supplementary Material Section D, we perform targeted ablations on CMU and DS232 cognitive-task datasets to assess each component’s role. Omitting QR decomposition produces embeddings that reflect overall data structure but lack subject-level discrimination, whereas removing LSH leaves subspace rotations uncorrected, impairing alignment consistency. Only the full OCL model—jointly leveraging QR and LSH—delivers robust, discriminative embeddings across subjects and sites, demonstrating that both modules are indispensable to our framework.
>
> 2. Computational Overhead
>
> We have quantified OCL’s computational demands on two workstations—each outfitted with dual NVIDIA GeForce RTX 4060 Ti GPUs (16 GB VRAM). As detailed in Supplementary Material Section C and shown in Figure S1, OCL’s pretraining and fine-tuning runtimes are on par with leading self-supervised fMRI frameworks (e.g., MindEye2, MindAligner) and mirror the memory footprints of CLIP-based and other Transformer models adapted for neuroimaging. End-to-end inference on typical fMRI datasets requires approximately 14 GB of VRAM on a single GPU, confirming that OCL’s resource requirements are moderate and practical for the broader neuroimaging community. In the revised manuscript, we will extend this discussion by analyzing how memory usage and throughput scale with voxel count, time-series length, and model depth.
>
> 3. Suitability for fMRI
>
> Our pretrained synthetic-data approach effectively mitigates sample scarcity in small fMRI datasets, enabling OCL to generalize from limited real data. The architecture is rooted in hyperalignment theory—QR decomposition aligns subject-specific neural subspaces, and LSH incorporates robustness to the low SNR characteristic of fMRI signals. Together, these components are specifically tailored for noisy neuroimaging data, and our empirical results across multiple benchmarks provide compelling evidence of OCL’s suitability and resilience in real-world fMRI analysis.
>
> 4. Role of Positional Encoding
>
> We employ sinusoidal positional encoding—widely adopted in prior fMRI Transformer models—to inject continuous temporal context into spatial feature representations. Although the encoding formula is standard, our contribution lies in integrating it with QR-derived bases and LSH signatures under variable-length padding masks, ensuring that temporal ordering is preserved even when time-series lengths differ. This integration is critical for aligning dynamic stimulus responses without explicit preprocessing and is not a direct copy but an adaptation tailored to OCL’s multi-component layers.
>
> 5. Limitations and Dual-Use Considerations
>
> We will add a dedicated Limitations section in the final manuscript to acknowledge:
>
> Societal & Dual-Use Risks: OCL could be applied to sensitive tasks (e.g., cognitive state inference or mental health diagnostics). We will recommend ethical safeguards, strict anonymization, and compliance with institutional review boards to mitigate misuse.
>
> Scalability Boundaries: OCL’s multi-module architecture demands multi-GPU hardware (≈14 GB VRAM per GPU); we will discuss optional reductions in embedding dimension or Transformer depth to trade off accuracy and resource use.
>
> Data Requirements & Generalization: OCL benefits from large, well-annotated multi-site datasets; we will caution about performance degradation under low-data regimes and suggest few-shot adaptation or synthetic augmentation as remedies.
>
> 6. Presentation & Reproducibility
>
> We apologize that presenting results solely with bar charts may limit clarity and reproducibility. While visual summaries provide quick insights, we will include detailed numerical tables in the Supplementary Material to ensure full traceability and precise comparisons.
>
> 7. Task Scope & Future Directions
>
> Our primary objective is to deliver a predictive alignment framework with immediate practical utility for classification-based fMRI analyses in clinical and cognitive neuroscience. Although more advanced brain-decoding tasks—such as retrieval or stimulus reconstruction—offer valuable insights, they extend beyond the current scope of this paper. Crucially, OCL already surpasses recent CLIP-based and other Transformer models (e.g., MindEye2 or MindAligner) in classification accuracy across multi-site benchmarks, highlighting its efficacy in real-world predictive scenarios. We plan to investigate retrieval and reconstruction tasks in future work, leveraging OCL-aligned representations as a foundation.
>
> Thank you again for your insightful feedback. We hope this revision addresses all your concerns and highlights OCL’s practical benefits and scientific rigor. We appreciate your consideration and kindly ask for your endorsement of our manuscript.
>
>
> Sincerely,
>
> The Author(s)

---

> > ### Comment · Reviewer_xXsd · 2025-08-04
> >
> > Thanks for your reply, I have carefully read through your feedback. Your reply has solved most of my doubts.
> >
> > You have addressed the two primary concerns I raised: the ablation studies demonstrating the necessity of QR decomposition and LSH, and the analysis of computational overhead associated with the method’s complexity. This further strengthens my belief in the value of the proposed work for multi-subject and multi-site fMRI classification tasks. However, the paper still lacks comparison with more recent, advanced multi-subject decoding models, and does not explore other important decoding tasks such as retrieval and reconstruction.
> >
> > I will revise my rating (from 3 to 4) . Look forward to the code and model weights going open source. Good luck！

---

### Official Review · Reviewer_aFC5 · 2025-07-02

**Clarity:** 4
**Significance:** 3
**Originality:** 3
**Rating:** 5
**Confidence:** 3

**Summary:**

This paper introduces **Orthogonal Contrastive Learning** (OCL). The method aligns fMRI time-series across subjects and scanning sites without any temporal preprocessing. OCL employs a dual-encoder setup and extracts orthogonal features using QR decomposition. It then generates compact signatures via locality-sensitive hashing and injects time information continuously before embedding with a Transformer. The framework is pretrained on synthetic data and uses a cross-site transfer-learning workflow to boost downstream classification accuracy.

**Questions:**

1. **Computational Complexity:** Could the authors clarify the computational overhead associated with integrating multiple sophisticated modules (QR decomposition, LSH, positional encoding, Transformer)? Providing runtime or memory benchmarks would significantly strengthen the practical applicability of the proposed method.

2. **Ablation Studies:** Could the authors present further ablations to individually highlight each component’s contribution (QR decomposition, LSH, positional encoding, Transformer)? This would greatly help readers understand the relative importance of each design choice.

3. **Generalizability to Other Modalities:** Have the authors explored or considered applying OCL to resting-state fMRI or other neuroimaging modalities? Demonstrating broader applicability would significantly increase the potential impact of the work.

**Ethical Concerns:**

["NO or VERY MINOR ethics concerns only"]

**Final Justification:**

The rebuttal provided by the authors addresses my major concern. And I'd like to decide my final rating as **Accept**.

**Limitations:**

**No.**

While the paper has a “Broader Impacts” section that describes its positive outcomes, it does not have a clear discussion of its own limitations. It also does not consider possible negative effects or dual‐use risks.

**Suggestions:**

- **Computational cost & scalability** OCL uses two encoders and deep transformer layers. This may need large GPU resources. Smaller labs may find it hard to run.
- **Data requirements & generalization** The method needs big, well‐labeled fMRI datasets from many sites. This may not work for under‐represented groups or new tasks.
- **Assumption sensitivity** The alignment assumes orthogonality and temporal stationarity. The authors should discuss what happens if these assumptions do not hold.

**Quality:**

3

**Strengths And Weaknesses:**

**Strengths:**
1. The OCL framework addresses a significant practical challenge—aligning heterogeneous fMRI datasets without requiring temporal preprocessing, potentially reducing overhead and improving flexibility.
2. Experimental results appear solid, demonstrating improvements in alignment quality and downstream classification accuracy compared to several state-of-the-art methods across multiple multi-site benchmarks.
3. Design choices, such as combining QR decomposition for orthogonal feature extraction with LSH signatures, seem well-motivated and original. Using Transformers for temporal dependencies also seems intuitive and empirically effective.

**Weaknesses:**
1. The method integrates several complex modules (QR decomposition, LSH, positional encoding, Transformer layers), which might increase computational complexity. A more detailed discussion on scalability or efficiency would be beneficial.
2. Ablation studies could be expanded. Currently, the relative contributions of individual components like QR decomposition, LSH, positional encoding, or Transformer layers are not fully clear. Further clarification here would strengthen the paper’s claims.

---

> ### Author Rebuttal · Authors · 2025-07-31
>
> Dear Reviewer,
>
> Thank you for your positive evaluation and for highlighting both the strengths of OCL and areas for clarification. Below, we address each of your points in detail.
>
> 1. Computational Complexity & Scalability
>
> We have quantified OCL’s overall computational overhead both analytically and empirically. As noted in Section 4, experiments were run on two workstations (four NVIDIA GeForce RTX 4060 Ti GPUs total), with end-to-end inference requiring about 14 GB of VRAM on a single GPU. Further, runtime analysis is provided in Supplementary Material Section C. In the revised manuscript, we will extend this discussion by examining how resource usage and throughput scale with voxel count, time-series length, and model depth, and by offering practical guidelines for batch sizes and hardware configurations. These results demonstrate that OCL’s computational demands are moderate and comparable to other transformer-based neuroimaging methods.
>
> 2. Ablation Studies
>
> We agree that isolating each component’s contribution strengthens the interpretability of design choices. In Supplementary Material Section D, we report an ablation on two cognitive-task datasets (CMU and DS232), comparing four variants (removing QR, LSH, positional encoding, or the Transformer) against OCL_zero and random-initialized pipelines. Results (Figure S2) show that removing any component degrades accuracy significantly—confirming that QR, LSH, positional encoding, and the Transformer each play an essential role.
>
> 3. Generalizability to Other Modalities
>
> While our current evaluation focuses on task-based fMRI, OCL’s core mechanisms are inherently modality-agnostic, and we have actively considered applying it to resting-state and other neuroimaging modalities. For resting-state fMRI, pseudo-labels can be generated via sliding-window functional connectivity or clustering of temporal segments, enabling OCL to learn contrastive alignment without explicit tasks. For MEG and EEG, OCL can process sensor- or source-space time–frequency matrices by extracting orthogonal spectral bases with QR decomposition, hashing them via LSH signatures, and preserving temporal dynamics through positional encoding. Although comprehensive empirical evaluation on these modalities is beyond the scope of this submission, we include a detailed “Future Extensions” subsection in the revised manuscript, outlining protocols and preliminary guidelines for applying and benchmarking OCL on resting-state fMRI, MEG, and other time-series neuroimaging data.
>
> 4.  Discussion of Limitations:
>
> In the Conclusion and Broader Impacts sections, we already address batch-effect mitigation, reproducibility considerations, and computational costs; in the revised manuscript, we will add a dedicated Limitations section that explicitly highlights (i) scalability constraints of OCL’s multi-module architecture on GPU resources, (ii) potential sensitivity to extreme inter-site or inter-subject domain shifts beyond our current benchmarks, and (iii) its reliance on large, well-annotated multi-site datasets for robust performance and generalization.
>
> We appreciate your support for our work and believe that these clarifications, analyses, and discussions will reinforce OCL’s practical applicability and scientific rigor. We would be grateful for your recommendation to accept our manuscript.
>
> Sincerely,
>
> The Author(s)

---

> > ### Comment · Reviewer_aFC5 · 2025-08-04
> >
> > Thanks the rebuttal provided by the authors. It addresses my major concerns.

---

### Official Review · Reviewer_ThRu · 2025-07-03

**Clarity:** 3
**Significance:** 4
**Originality:** 3
**Rating:** 4
**Confidence:** 4

**Summary:**

The paper introduces a novel method, OCL (Orthogonal Contrastive Learning), designed to enhance multi-subject and multi-site fMRI data analysis. The authors provide a thorough theoretical foundation and empirical validation of their approach, demonstrating improvements in feature representation quality and classification accuracy compared to existing methods.

**Questions:**

1.Baseline Comparisons: Lacks implementation details and hyperparameter settings for baselines like DeepGeoCCA and MindAligner, raising concerns about the fairness of comparisons.
2.Generalizability: All experiments are on task-based fMRI; applicability to resting-state data or other modalities (e.g., MEG, EEG) is not evaluated or discussed.
3.Computational Complexity: OCL uses deep transformers and large-scale synthetic pretraining, but memory usage, scalability, and training efficiency are not analyzed.
4.LSH Robustness: While LSH is central to OCL, its sensitivity to noise, outliers, or small samples is not empirically validated.
5.Interpretability: Despite strong classification performance, the paper lacks neuroscientific interpretation or visualization of learned features.
6.Synthetic Data Realism: Pretraining uses simplistic Gaussian models that may not reflect real BOLD signal complexity, with no assessment of realism or potential bias.

**Ethical Concerns:**

["NO or VERY MINOR ethics concerns only"]

**Final Justification:**

I am still positive to this paper.

**Limitations:**

refer to the questions

**Quality:**

3

**Strengths And Weaknesses:**

Strengths
1.Innovative Framework: OCL combines orthogonal feature extraction, contrastive learning, and transformers to tackle key challenges in multi-subject fMRI analysis.
2.Well-Designed Architecture: Uses a dual-encoder framework with online and target networks, integrating QR decomposition, LSH, positional encoding, and transformers effectively.
3.Strong Performance: Demonstrates consistent superiority over existing methods across multiple fMRI datasets in both representation quality and classification accuracy.
4.Comprehensive Evaluation: Tested on diverse fMRI tasks (task-based and naturalistic), with rigorous metrics, statistical analysis, and comparisons under various conditions.

This paper presents a promising and original approach for multi-subject and multi-site fMRI analysis. Addressing the points above would greatly strengthen the contribution and increase its relevance to both machine learning and neuroscience audiences.

---

> ### Author Rebuttal · Authors · 2025-07-31
>
> Dear Reviewer,
>
> Thank you for your thorough and constructive feedback on our OCL manuscript. We appreciate your recognition of the framework’s technical soundness and originality. Below, we address each of your points in detail.
>
> 1. Baseline Comparisons
>
> We have tuned all competing methods (DeepGeoCCA, MindAligner, etc.) through nested leave-one-subject-out cross-validation, performing grid search on hyperparameters (kernel scales, regularization coefficients, learning rates) in the same way as for OCL. Implementations and pretrained weights follow the original papers, and we confirmed parameter choices with the authors where necessary. We evaluated each baseline at both fixed and optimally chosen latent dimensions, reporting the best results.
>
> 2. Generalizability
>
> OCL is fundamentally modality-agnostic: the QR decomposition extracts orthogonal bases from any time-series matrix, and the LSH signature encodes these bases into compact, comparable codes. For resting-state fMRI, one can generate pseudo-labels by applying sliding-window functional connectivity or clustering temporal windows based on correlation patterns; OCL then maximizes contrastive agreement across these pseudo-classes, aligning subjects without explicit tasks. Similarly, for MEG/EEG data, the decomposition can operate on sensor- or source-space time-series, and LSH can hash spectral features or time–frequency representations. Positional encoding preserves temporal ordering even when sampling rates vary between modalities. Although detailed experiments on these extensions exceed our current scope, we include a dedicated “Future Extensions” discussion highlighting how to adapt OCL’s preprocessing and contrastive objectives to new neuroimaging modalities.
>
> 3. Computational Complexity
>
> Experiments were conducted on two workstations (four GPUs total), each equipped with two NVIDIA GeForce RTX 4060 Ti GPUs (16 GB VRAM each). Supplementary Material Section C reports runtime footprints. For typical datasets, OCL inference requires approximately 14 GB VRAM on a single GPU. In the revised manuscript, we will expand this discussion by analyzing how memory usage and throughput scale with voxel count, time-series length, and model depth; we will also include guidelines for selecting batch sizes and hardware configurations for different dataset scales. The current analyses confirm that OCL’s resource demands are moderate and comparable to other transformer-based neuroimaging approaches.
>
> 4. LSH Robustness
>
> Our theoretical analysis (Lemma 1) shows that LSH collision probability decreases smoothly with increasing Euclidean distance between orthogonal codes, providing inherent resilience to noise and minor perturbations. To validate this, we conducted a new empirical study in which we injected Gaussian noise into the synthetic pretraining data at SNR levels of 10 dB, 5 dB, and 0 dB. Across all conditions, classification accuracy dropped by less than 1 %, demonstrating high robustness. We will include these new analyses, along with additional experiments on outlier sensitivity and varying sample sizes, in the final version of the manuscript
>
> 5. Interpretability
>
> We acknowledge that interpretability in deep-learning–based neuroimaging is challenging and not the primary focus of this work, which aims to advance predictive alignment. Nevertheless, to provide initial insights into learned representations, we project the QR bases back into voxel space and visualize the most significant spatial components. Additionally, we extract self‑attention weights from the Transformer encoder and overlay these attention maps onto cortical surfaces to indicate which temporal segments drive alignment. We will include a discussion of these visualizations and their neuroscientific implications in the revised manuscript.
>
> 6. Synthetic Data Realism
>
> The Gaussian model in our pretraining is intended to teach orthogonal invariance rather than fully emulate hemodynamics. On held‑out real data, OCL representations improve both classification and within‑class correlation (ρ_raw=0.763→ρ_aligned=0.915, please see Supplementary Material Section B), indicating no adverse bias. We will acknowledge the value of more realistic simulators (e.g., HRF modelling) in future work.
>
> We trust these clarifications address your concerns and strengthen our contribution. Thank you again for your valuable feedback. We appreciate your consideration and hope you will endorse our manuscript for acceptance.
>
> Sincerely,
>
> The Author(s)

---

### Note · Authors · 2025-08-12

We sincerely thank the Area Chair and reviewers for their careful evaluation and constructive feedback. Your comments helped us clarify our motivation, tighten the experiments, and improve the presentation of Orthogonal Contrastive Learning (OCL). Below is a brief summary of the key final points:

1. Purpose-built design: OCL is natively engineered for fMRI—rather than adapted from general pretrained models—combining orthogonal alignment (QR), subject signatures (LSH), continuous temporal encoding, and a Transformer to align variable-length time series without explicit temporal preprocessing. In the final version, we will also discuss how OCL can potentially extend to other neuroimaging modalities - e.g., EEG, MEG.

2. Expanded empirical support: We report consistent gains across datasets, ablations confirming each module’s contribution, and multi-site evaluations demonstrating robust cross-site generalization. Additional empirical studies referenced in the rebuttal will be incorporated to further support OCL’s robustness.

3. Practical compute profile: We provide measured runtime/memory footprints indicating moderate, comparable resource needs relative to transformer-based neuroimaging pipelines. We will extend this discussion with clearer hardware configuration details and guidance on inference resource requirements.

4. Improved reproducibility and balance: We add concise numerical tables alongside figures, clarify ROI and experimental setup details, and include an explicit Limitations section (scalability, data requirements, modelling assumptions, and ethics).

We believe these updates address the reviewers’ concerns and show that OCL is both theoretically grounded and practically effective. We hope you will endorse our manuscript for publication.

Thank you again for your time and thoughtful evaluation.

Sincerely,

The Author(s)

---

### Decision · Program_Chairs · 2025-09-17

**Decision:**

Accept (poster)

**Comment:**

This paper introduces Orthogonal Contrastive Learning (OCL), a representation framework designed for task-based fMRI analysis involving multiple subjects and sites. This method incorporates QR decomposition, locality-sensitive hashing, positional encoding, and a transformer encoder within a dual-encoder architecture comprising online and target networks. The model is trained using contrastive loss to align same-stimulus fMRI responses across subjects while preserving distinctiveness across different stimuli. Pretraining on synthetic fMRI-like data provides robust initialization, followed by fine-tuning on real data. Extensive experiments demonstrate that OCL outperforms various state-of-the-art functional alignment and contrastive learning methods in terms of both representation quality and downstream classification performance.
The reviewers noted that the paper is more oriented toward the technique than toward applications.
They raised questions about the baseline comparison, the need for ablation studies, the computational complexity, the robustness of the locality-sensitive hashing (LSH) encoding, the realism of the synthetic data, and the generalization to other modalities and interpretability.
These points have been successfully addressed in the rebuttal process. However, this has resulted in a significant discrepancy between the submitted paper and the intended final version. Other concerns remain, such as the lack of consideration of spatiotemporal information in fMRI data. However, the consensus is that the paper is acceptable.